# U.S. Federal Investment in Religiousness/Spirituality and Health Research: A Systematic Review

Crystal L. Park [1,*] , Jamilah R. George [1], Saya Awao [1], Lauren M. Carney [1], Steven Batt [2] and John M. Salsman [3]

1   Department of Psychological Sciences, University of Connecticut, Storrs, CT 06269, USA
2   University of Connecticut Library, Storrs, CT 06269-1005, USA
3   Wake Forest University School of Medicine, Winston-Salem, NC 27101, USA
*   Correspondence: crystal.park@uconn.edu

**Abstract: Objectives:** Although robust associations between aspects of religiousness/spirituality (R/S) and physical health have been established, little systematic information is available about federal funding support for this area of research. To address this question, we conducted a comprehensive systematic review and analysis. **Study Design:** Systematic review. **Methods:** We used the information provided by the Federal RePORTER and searched from earliest date through the end of 2018. Abstracts were included if they were an empirical study and included both a religion/spirituality variable and a health variable. **Results:** Our search yielded 194 grants reflecting over USD 214 million in research expenditures, with the vast majority (85%) funded by the NIH. Most common were community-based observational studies with healthy populations (70%). Nearly three-quarters (73%) of studies specifying age focused on adults, but children and adolescents were also well represented in these projects. The proportion of studies focused on racial/ethnic minorities (47%) was disproportionate to their representation in the U.S. population, which could reflect either heightened efforts to address health disparities or a view that R/S is primarily or mostly relevant to minority groups. Less than half of funded studies (41%) considered religion a central focus and publications for R/S-focused studies were less common than for non-R/S-focused studies (M = 7.0 to M = 13.3, respectively, $p = 0.06$). Overall funding levels appear to be declining in more recent years, although this trend was not statistically significant ($p = 0.52$). Many abstracts did not provide adequate details for coding. **Conclusions:** Overall, the present review suggests that U.S. federal funding for research on R/S and health is substantial, but most of this has only peripherally considered R/S and has yielded modest return on investment. Promising future directions include a continued focus on racial and ethnic minority populations as well as in emerging areas such as religious gratitude and compassion along with well-designed intervention trials.

**Keywords:** religion; health; federal funding; systematic review

## 1. Introduction

Among the many psychosocial aspects that may influence health, religiousness/spirituality (R/S) tends to receive relatively little mainstream attention. This lack of attention may derive from the general skepticism voiced by many social scientists regarding R/S (Rutjens and Preston 2020) or the reluctance expressed by many healthcare providers regarding engaging in patients' spiritual concerns (Greenblum and Hubbard 2019). Yet R/S is an important aspect of many people's lives (Park et al. 2016), and many studies have established associations between R/S and physical health. The strongest evidence of this link is the robust association between religious service attendance and mortality (see (VanderWeele 2017) for a review). Less conclusive but growing evidence supports associations of attendance with additional health indicators, including cardiovascular disease (VanderWeele 2017). Other aspects of R/S, such as religious social support and religious coping (e.g.,

prayer), have received less attention but have also demonstrated associations with health (see (Park and Slattery Forthcoming) for a review).

However, the quality of research on R/S and health has generally been low. For example, most studies have employed less rigorous research designs (e.g., cross-sectional studies without a comparison group, observational studies with convenience samples) and have failed to include or control for confounding variables that may account for significant variance in outcomes (e.g., physical health status when predicting the association between religious service attendance and physical quality of life) (Park et al. 2016). Although all methods have strengths and limitations, the relatively low quality of this body of work as a whole has, in part, been considered by some as a byproduct of the relative neglect or bias against federally funded research on the topic of RS and health (see (Levin 2005; Levin 2016a)). Indeed, few federal health-related agencies highlight or even mention R/S in their materials, including on their myriad webpages. In addition, some researchers have noted the fraught legal issues involved in funding R/S work, raising the specter of mingling church and state (Levin 2016b), which may further hinder research in the field.

To date, no empirical research has directly evaluated the extent to which R/S-health research has received federal funding support. It may be that little federal funding is available or that it is primarily directed towards research with highly specific groups, such as religious communities, minorities, or people with specific conditions such as substance misuse (Levin 2016a). Another possibility is that federally funded investigators of R/S and health fail to produce substantive results. Thus, it is important to establish the investment of federal funding of R/S and health research—along with the specific areas in which these investments have been made and the publications that result from them.

To address this knowledge gap and guided by a "best practices" approach (Siddaway et al. 2019), we conducted a comprehensive systematic review and analysis of U.S. federally funded applications of RS and health. We used the Federal RePORTER, the most comprehensive database of federally funded studies available, as our source data. The Federal RePORTER includes, for example, funding from the U.S. Department of Veteran Affairs (VA), the Department of Defense (DOD), the National Science Foundation (NSF), and the National Institutes of Health (NIH).

We asked three questions to characterize the state of funding on R/S and health over time and its impact: (1) What are the methodological and sample characteristics of studies that received funding? In addition, we examined the extent to which an emerging area of importance, religiously relevant virtues (Carlisle and Tsang 2013; Krause and Hayward 2015), has been covered in these funded projects. (2) What is the overall funding portfolio? (3) What is the return on investment from funded projects? By exploring these questions, this review aims to provide an informative and data-driven perspective on the nature of funding to support R/S and health research and the relative degree to which particular areas of inquiry have been successful.

## 2. Method

This systematic review, using data provided by the Federal RePORTER, was exempt from Institutional Review Board review. Federal RePORTER was used to identify funded projects with an explicit focus on or inclusion of R/S variable(s). This systematic review adhered to recommended guidelines for the Preferred Reporting Items for Systematic Reviews and Meta-Analyses (PRISMA (Page et al. 2021)). Our planned review approach was described as part of our funded grant proposal and is available on request.

### 2.1. Search Strategy

A health science librarian conducted the search on Federal RePORTER using search terms for R/S (spiritual OR spirituality OR religio* OR faith OR prayer OR prayers OR praying OR church OR churches OR churchgoing OR temple OR temples OR synagogue OR synagogues OR mosque OR mosques OR Jewish OR Jews OR Christians OR Protestant OR Protestants OR Catholic OR Catholics OR Muslim OR Muslims OR Ramadan OR Mormon

OR Mormons OR Adventist OR Adventists OR Buddhist OR Buddhists OR Buddhism OR Hindi OR Hindus OR Hinduism OR Scientology OR Scientologists OR Jehovah OR Evangelical OR Evangelicals OR Baptist OR Baptists OR Pentecostal OR Pentecostals OR Taoist OR Taoism OR Sikhism OR Sufism) and narrowed the results by including "and health". We wanted to provide as broad of a search strategy as possible and so we anticipated that "health" would be sufficiently comprehensive to include outcomes focused on both physical and mental health, and we did not limit our search terms to the title or abstract only. What that means is that even if a project does not specifically reference health in the title or abstract, if it had a mental health outcome (e.g., depression, anxiety), then these outcomes would have been indexed as part of the project terms and thus captured by our search strategy. There was no lower date limit and we searched through 2018. All resulting grant abstracts were uploaded to Covidence (www.covidence.org, accessed on 20 March 2018), a web-based platform and non-profit service working in partnership with the Cochrane Collaboration designed to improve production and use of systematic reviews for health and wellbeing.

### 2.2. Eligibility Criteria

Projects were included if they (1) were an empirical research study, (2) included an R/S variable, (3) included a health outcome, and (4) examined the link between an R/S variable and a health outcome. Projects were excluded if they (1) were duplicate projects or (2) conducted in a religious setting or with a religious group but did not evaluate the role of R/S in health.

Two research assistants independently reviewed each abstract, rating whether it met eligibility criteria (yes/maybe/no). Abstracts that had conflicts (i.e., yes/maybe vs. no) were examined by a third rater (C.L.P. or J.M.S.) to resolve the disagreement. Abstracts with a yes or maybe consensus recommendation were next examined independently by two raters (C.L.P., J.M.S.) for final recommendation (include/exclude) with disagreements resolved by consensus. The review is registered on Open Science.

### 2.3. Data Coding

A codebook was developed to guide the independent review and data extraction process. Data were extracted directly from Federal RePORTER by two raters and recorded in Covidence. Discrepancies were resolved by consensus with larger group discussion as needed. Study demographics focused on age group, sex, race/ethnicity, country where the research was conducted, study type (observational vs. interventional), sample (healthy/community vs. disease/clinical), and health outcomes (physical, emotional, social, and spiritual).

R/S variables were coded for centrality, role, and construct type. Centrality of R/S was coded as "yes" if R/S was a key variable in the study (e.g., primary outcome) or as "no" if it was not a main focus of the study (e.g., one of many potential moderators). R/S construct type was coded as affective (e.g., spiritual well-being), behavioral (e.g., religious coping), cognitive (e.g., beliefs about God), or other (e.g., religious social support) (cf. (Salsman et al. 2015)). The specific roles of R/S variables in the studies were coded as predictor, outcome, mediator, moderator, or multiple categories.

To examine the extent to which religious-relevant virtues were included in the set of funded projects, we searched the included project titles, abstracts, and keywords for the terms "gratitude", "grateful", "humility", "humble", "forgiveness", "forgive", and "compassion". To further characterize the overall research portfolio, funded projects were coded by total amount funded, years of funding, funding agency, grant mechanism, number of publications, and type of research institution. The Carnegie Classification System of Institutions of Higher Education (Indiana University Center for Postsecondary Research n.d.) was used to characterize institution type (doctoral universities, master's colleges and universities, special focus institutions, baccalaureate colleges, and not applicable (N/A)). An institution was classified as N/A if it was not an institution of higher education

(e.g., National Bureau of Economic Research) or if it was an institution outside of the United States.

Lastly, to evaluate return on investment, we recorded the total number of publications for each project from PubMed using the grant numbers. This information is only available for NIH grants because only NIH requires reporting of publications linked to the grant. Because the R series (i.e., investigator-initiated) is the most common funding mechanism used by NIH, we compared the total number of publications for which R/S was central to those for which R/S was not central among R series grants. Abstracts of each publication were exported and examined to determine if R/S was highlighted in the publication. If a key term (e.g., religion, spirituality, church, prayer, etc.) was included in the abstract, R/S was considered a central component of the publication, and the DOIs, journal names, and number of citations were recorded. The impact factor of each journal was retrieved from Journal Citation Reports (clarivate.com). For 10 journals, impact factors were not available on this database. Of those, 9 journal impact factors were retrieved from another source (e.g., journal website) and 1 journal impact factor was not found from any source.

## 3. Data Analysis

Study sample demographics, R/S characteristics, health outcomes, funding institutes, and grant mechanisms were tabulated and summarized using frequency distributions. In addition, the mean, median, and standard deviation of funded grants were calculated per award and per year. Overall amount of funding by research dollars and awards were calculated and examined using linear regression to identify significant changes over time. Next, the average amount of funding and number of publications per grant were calculated by R/S centrality (yes/no) and compared using T-tests to identify significant differences. Finally, we tallied the number/percent of our set of religious-relevant virtues mentioned in the abstract/keyword/search materials.

## 4. Results

### 4.1. Study Selection

The Federal RePORTER database search retrieved 7532 grants (Figure 1). After removal of duplicates, 7523 remained and were initially screened to ensure inclusion of a religious or spiritual variable. Of these, 6391 clearly did not meet the inclusion criteria and were discarded, and 1132 were assessed further to ensure that all inclusion criteria were met. Of these, 938 did not meet the inclusion criteria, while 194 did and were included in subsequent analyses.

### 4.2. Characteristics of Funded R/S and Health Projects

Abstracts from funded grants varied in the details provided about the sample characteristics, research design, and measures used. Although many characteristics were not specified, there were some notable features of the funded studies. More than one-third (39%; k = 75/194) of the studies focused on adults (or 73% of studies that specified an age group (k = 75/103)). Similarly, 47% (92/194) of studies specifically included non-Hispanic white participants, and 32% (k = 62/194) and 15% (k = 29/194) of studies recruited Black/African American or Hispanic or Latino samples, respectively, corresponding to 53% (k = 62/118) and 25% (k = 29/118) of studies that described a specific racial or ethnic group. The majority of projects utilized samples from the United States (87%, k = 169/194), focused on community-based or otherwise healthy participants (70%, k = 136/194), and used an observational approach (72%, 139/194). R/S was central in only 41% of projects (k = 79/194), was typically used as a predictor variable (60%, k = 117/194), and most commonly focused on "other" dimensions of the R/S construct (45%, k = 88/194). In terms of the religious-relevant virtues, we found 0 projects mentioning gratitude/grateful, humility/humble, or forgiveness/forgive. Two projects included the term compassion. Not surprisingly, physical health outcomes were most commonly reflected in funded projects (61%, k = 119/194),

but emotional (57%, 110/194) and spiritual health outcomes (52%, 101/194) were also common. (See Table 1 for additional characteristics of funded projects.)

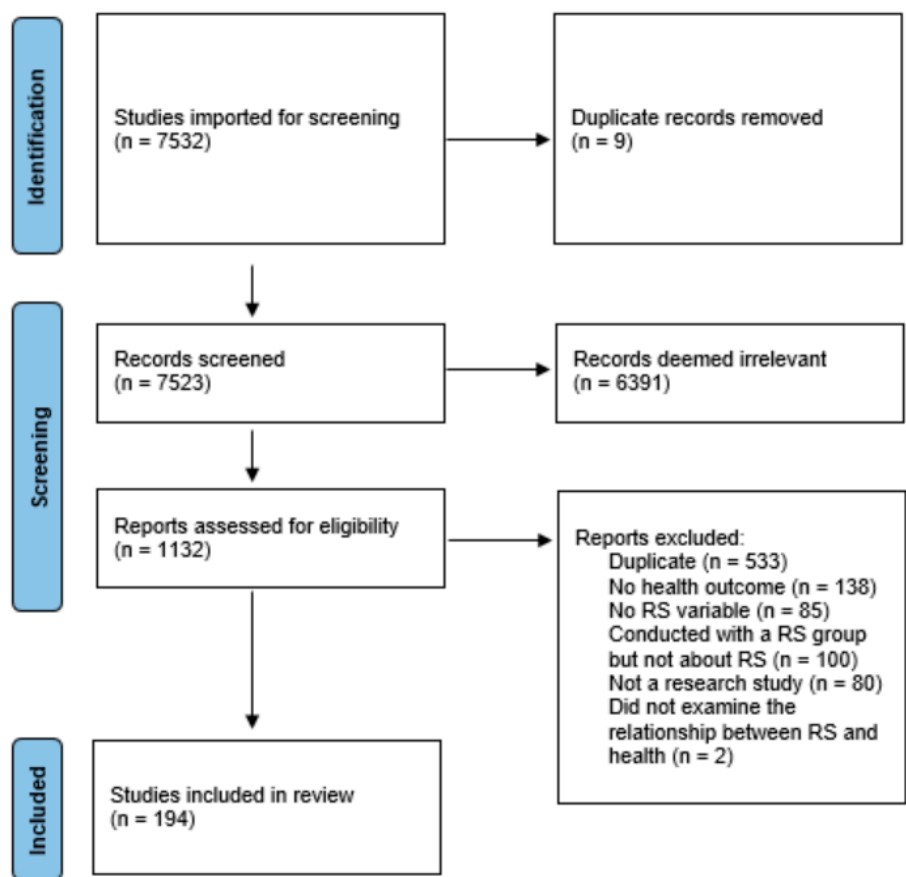

**Figure 1.** PRISMA flow diagram.

**Table 1.** Proposed study demographics and R/S characteristics (k = 194).

| Age Group | N | % |
|---|---|---|
| Older Adults | 14 | 7.2 |
| Adults (non-specified) | 57 | 29.4 |
| Young Adults | 4 | 2.1 |
| Pediatric | 28 | 14.4 |
| Unspecified | 103 | 53.1 |
| **Sex** | | |
| Female | 48 | 24.7 |
| Male | 31 | 16.0 |
| Unspecified | 134 | 69.1 |
| **Race/Ethnicity** | | |
| White | 12 | 6.2 |
| Black/African American | 62 | 32.0 |
| Asian or Pacific Islander | 9 | 4.6 |
| Native American or Alaskan Native | 6 | 3.1 |
| Native Hawaiian or Other Pacific Islander | 0 | 0.0 |

**Table 1.** *Cont.*

| | | |
|---|---|---|
| Hispanic or Latino | 29 | 14.9 |
| Unspecified | 101 | 52.1 |
| **Country** | | |
| United States | 169 | 87.1 |
| International | 31 | 16.0 |
| **Sample Type** | | |
| Healthy/Community | 136 | 70.1 |
| Disease: Cancer | 16 | 8.2 |
| Disease: Psychiatric disorder | 13 | 6.7 |
| Disease: Other (e.g., asthma, infertility, obesity) | 10 | 5.1 |
| Disease: HIV/AIDS | 9 | 4.6 |
| Disease: Multiple chronic conditions | 6 | 3.1 |
| Disease: Cardiovascular | 4 | 2.1 |
| **Study Type** | | |
| Observational | 139 | 71.6 |
| Interventional | 52 | 26.8 |
| Unspecified | 3 | 1.5 |
| **R/S Centrality** | | |
| Yes | 79 | 40.7 |
| No | 115 | 59.3 |
| **R/S Variable Role** | | |
| Predictor | 117 | 60.3 |
| Outcome | 54 | 27.8 |
| Mediator | 23 | 11.9 |
| Moderator | 25 | 12.9 |
| Multiple | 33 | 17.0 |
| **R/S Construct** | | |
| Affective | 38 | 19.6 |
| Behavioral | 51 | 26.3 |
| Cognitive | 29 | 14.9 |
| Other | 88 | 45.3 |
| Unspecified | 8 | 4.1 |
| **Health Outcomes** | | |
| Physical | 119 | 61.3 |
| Emotional | 110 | 56.7 |
| Social | 75 | 38.7 |
| Spiritual | 101 | 52.1 |

Note: Cells do not always add to 100% within categories as some characteristics are reflected in multiple projects.

*4.3. Funding Portfolio of R/S and Health Research*

The total amount funded across all 194 grants was USD 214,171,986. Per grant, the mean was USD 1,145,305 (SD = USD 1,549,018), the median was USD 577,665 and the range was USD 4496–12,974,479. Average number of years funded was 3.4 years (SD = 1.6 years). The majority of awards went to doctoral universities designated as "very high research

activity" (53%, 102/194) with those designated as "high research activity" (5%, 10/194) and as "professional universities" (0.5%, 1/194) also receiving a modest number of awards. Four-year medical schools and centers received 28 awards (14%) and master's colleges and universities designated as "larger programs" or "small programs" received 8 (4%) and 1 (0.5%) award(s), respectively. Forty-four awards (23%) were made to institutions not rated by the Carnegie Classification system.

Nine of the top ten funding agencies by number of awards granted were from NIH, which accounted for 85% of all funded projects (k = 164/194). NSF was the only federal agency not part of the NIH that was represented in the top ten funders, awarding an additional 17 projects. Within the NIH, the National Cancer Institute and the National Institute on Aging provided the largest investments with USD 41,482,047 and USD 31,639,406 in awarded research support, respectively. Table 2 provides additional details by funding agency.

**Table 2.** Number of grant awards by federal agency.

| Agency | Awards | Amounts (USD) |
| --- | --- | --- |
| * National Institute of Child Health and Human Development | 26 | 27,438,617 |
| * National Cancer Institute | 22 | 41,482,047 |
| * National Institute of Mental Health | 22 | 20,860,171 |
| National Science Foundation | 17 | 604,771 |
| * National Institute on Minority Health and Health Disparities | 17 | 19,510,115 |
| * National Institute of Nursing Research | 16 | 20,624,542 |
| * National Institute on Drug Abuse | 14 | 11,615,678 |
| * National Institute on Aging | 12 | 31,639,406 |
| * National Institute on Alcohol Abuse and Alcoholism | 12 | 11,342,277 |
| * National Heart, Lung, and Blood Institute | 9 | 19,306,572 |
| * National Center for Research Resources | 5 | 330,131 |
| U.S. Department of Veterans Affairs | 4 | unavailable |
| * National Center for Complementary and Integrative Health | 3 | 1,245,133 |
| National Institute of Food and Agriculture | 3 | unavailable |
| * Fogarty International Center at NIH | 2 | 707,235 |
| National Center for HIV/AIDS, Viral Hepatitis, STD, and TB Prevention | 2 | 1,615,521 |
| * National Institute of Dental and Craniofacial Research | 2 | 735,000 |
| * National Institute of General Medical Sciences | 2 | 1,471,543 |
| Agency for Healthcare Research and Quality | 1 | 41,510 |
| Congressionally Directed Medical Research Programs | 1 | 1,177,582 |
| U.S. Department of Health and Human Services | 1 | 1,375,163 |
| National Center for Injury Prevention and Control | 1 | 1,048,972 |
| Total | 194 | 214,171,986 |

* Indicates a federal agency within the National Institutes of Health.

Not surprisingly, among NIH grants, R series grants were the most common awards (53%, k = 102/194), with the R01 mechanism accounting for almost two-thirds of the R grant awards (65%, 64/99). (See Table 3 for funding by grant mechanism.) Lastly, we examined funding patterns over time. Figure 2 depicts trends in the number of R/S and health projects funded and the amount of funding over time. Of note, the full range of grants funded was not available in the Federal RePORTER prior to 2008, so awards reflected in 2003 to 2007 were projects that continued through 2008 or later. These years will underestimate the total number of projects awarded. Overall, from 2008 through 2018, there was a modest but non-significant decrease in the total amount of funding ($F$ (1,14) = 0.43, $p$ = 0.52) and number of projects ($F$ (1,14) = 0.13, $p$ = 0.72).

**Table 3.** Grant mechanism and type of R series grant for NIH-funded projects.

| Mechanism | Frequency | Percent |
| --- | --- | --- |
| R | 102 | 52.6 |
| R01 | 64 | 33.0 |
| R21 | 14 | 7.2 |
| R03 | 12 | 6.2 |
| R15 | 2 | 1.0 |
| R34 | 2 | 1.0 |
| R36 | 2 | 1.0 |
| R24 | 1 | 0.5 |
| R25 | 1 | 0.5 |
| R35 | 1 | 0.5 |
| R56 | 1 | 0.5 |
| RL1 | 1 | 0.5 |
| R00 | 1 | 0.5 |
| Missing | 22 | 11.3 |
| F | 18 | 9.3 |
| K | 17 | 8.8 |
| P | 12 | 6.2 |
| U | 8 | 4.1 |
| M | 5 | 2.6 |
| Z | 5 | 2.6 |
| I | 4 | 2.1 |
| SC | 1 | 0.5 |

R = research grants: R01 = NIH Research Project Grant Program, R21 = NIH Exploratory/Developmental Research Grant Award, R03 = NIH Small Grant Program, R15 = NIH Academic Research Enhancement Award, R34 = NIH Clinical Trial Planning Grant Program, R36 = Research Dissertation Award, R24 = Resource-Related Research Projects, R25 = Education Projects, R35 = Outstanding Investigator Award, R56 = NIH High Priority, Short-Term Project Award, RL1 = Linked Research Project Grant, R00 = NIH Pathway to Independence Award (K99/R00), F = Fellowship Programs, K = Research Career Programs, P = Research Program Projects and Centers, U = Cooperative Agreements, M = General Clinical Research Centers Program, Z = Intramural Projects, I = Non-HHS and Non-DHHS Federal Awards, SC = Research-Related Programs.

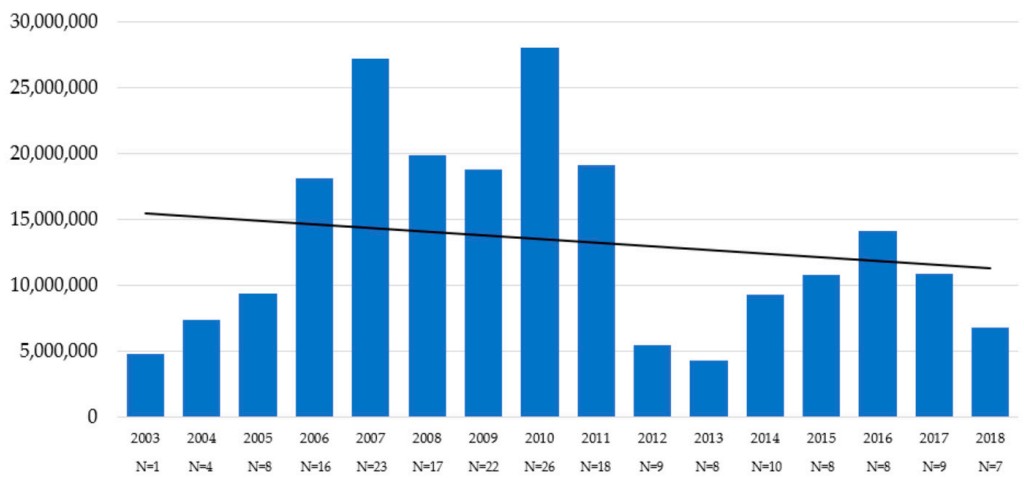

**Figure 2.** Total funding in USD of R/S projects by year.

*4.4. Return on Investment*

Analyses of funding and resultant publications for NIH-funded R series grants for which R/S was central (M = USD 1,352,306) received less funding compared to those that did not focus on R/S (M = USD 1,786,734), but these differences were not statistically significant ($p$ = 0.17). Notably, grants in which R/S was central had fewer publications (M = 7.0) compared to non-R/S focused grants (M = 13.3). However, these differences were only marginally statistically significant ($p$ = 0.06). Of the grants for which R/S was a central focus, almost two-thirds (63.8%) of their publications highlighted the role of R/S

and health, but the impact factor of those publications was modest (i.e., mean of 2.2) even though the average article citations were relatively strong (i.e., mean = 9.3).

## 5. Discussion

This review represents the first systematic inquiry into U.S. federal funding for research on R/S and health. Contrary to some widely held speculation, the federal government (especially the NIH) appears to have made a sizeable investment in research that includes a focus on R/S and health. This investment is spread across NIH institutes and other federal agencies. However, both the amount of money invested and the number of projects funded appear to be decreasing over time. This decline is occurring in the context of increasing budgets for federal research. For example, the NIH budget increased by approximately 30% in the years covered in the present review (Congressional Research Service 2021). In addition, R/S was a central component of less than half of the funded research reviewed here (41%); in the rest of the funded research, religion was one of many psychosocial variables studied.

The breadth and variety of these grants render characterizing prototypic studies of R/S and health difficult. Clearly, this research is not fully concentrated in specific groups. However, community-based research with healthy populations using an observational design is most common. Most of the studies specifying age focused on adults, but children and adolescents were also well-represented in these projects. The number of studies focused on racial/ethnic minorities is disproportionate to their representation in the U.S. population and could reflect either heightened efforts to address health disparities or a view that R/S is primarily or mostly relevant to minority groups (Holt et al. 2017). Regardless, given the increasing racial or ethnic diversity within the U.S. population (U.S. Census Bureau 2010), this focus may represent an important opportunity for sustained attention within R/S and health research. In addition, R/S is often conceptualized as a predictor variable of multiple aspects of physical and emotional health reflecting the larger body of research identifying significant associations (VanderWeele 2017; Park and Slattery Forthcoming; Jim et al. 2015; Lucchetti et al. 2021).

Return on investment appears to favor studies in which R/S is not central, as those studies have more publications associated with their grants (although this finding was merely a trend in terms of statistical reliability). Publications from R/S-focused grants that highlighted the role of R/S, and health in their manuscripts are typically published in journals with a modest impact factor (i.e., <5). This finding, however, should be interpreted with caution. A journal's impact factor is a useful but flawed index of the relative importance of the research (Smith 2006). For example, journals with more issues and articles can have higher impact factors. Similarly, the journal impact factor is not necessarily an indication of the value of a scientific article. The citation of the individual articles is a more reliable guide of impact. For context, with 10 or more citations, an article is typically in the top 24% of the most cited work worldwide (Van Noorden et al. 2014). Since the R/S focused articles were cited an average of 9.3 times, this suggests a more meaningful impact of the R/S findings than that suggested by impact factor alone.

While these findings offer key insights into the nature of U.S. federal funding for research on R/S and health, limitations of our review must be noted. While the funding for R/S and health research appears substantial, it is difficult to contextualize these support levels without information on federal investments in other types of psychosocial factors. Our ability to identify specific study elements was limited in that we had access only to the abstracts of the grants rather than the full proposals (which are not publicly available); specific aspects of the study were often described poorly or not at all. The most commonly studied type of R/S was classified as "other", which could be an amalgam of different types of R/S in a single measure or a single item reflecting religious affiliation or could reflect the lack of specificity or clarity provided in abstracts. Finally, while Federal RePORTER is broad in coverage, it does not include foundation funding, which might fund a higher percentage of R/S and health research (e.g., John Templeton Foundation).

Despite these limitations, the present systematic review suggests that the U.S. federal government has invested substantially in research on R/S and health. However, the majority of this research is limited—less than half of it has considered R/S a central aspect of the study, and much of it has been observational. Furthermore, the trends in funding appear to be in decline in more recent years. Given the increasing evidence that at least some aspects of R/S are strongly associated with health (VanderWeele 2017; Park and Slattery Forthcoming) (Lucchetti et al. 2021), it might behoove federal agencies to encourage and support applications in which R/S is a central component. In addition, very few of the funded projects (27%) involved an intervention; this field of research may be maturing to the point where R/S interventions are appropriate (e.g., counteracting misinformation and vaccine hesitancy among some R/S groups). Researchers funded to study R/S and health should be encouraged to continue publishing high quality work and to target high-impact journals to better disseminate their findings.

Another consideration for future federal grant-seekers is to continue to broaden the scope of the types of R/S focused on in the research. For example, religiously relevant virtues such as gratitude, humility, and forgiveness have emerged as important and under-researched aspects of R/S that may influence health (Krause and Hayward 2015). Gratitude to God has been shown to related to myriad aspects of mental and physical health (e.g., Krause 2009; Upenieks and Ford-Robertson 2022), yet our review turned up no federally funded research on this topic. Finally, the concentration of R/S and health research in minority populations needs to be better understood. Efforts to promote health specifically in underserved groups can be fostered through community-based participatory research strategies (e.g., (Culhane-Pera et al. 2021)), and R/S may be an important aspect of these efforts. R/S is important to many people across race, ethnicity, and socioeconomic status, and research on its associations with health should be explored in order to fully maximize health and wellbeing for all.

**Author Contributions:** Conceptualization, J.M.S. and C.L.P.; methodology, J.M.S. and C.L.P.; abstract search, S.B. and J.R.G.; data extraction, J.R.G. and S.A.; formal analysis, J.M.S. and L.M.C.; writing—original draft preparation, C.L.P., J.M.S., L.M.C. and S.A.; writing—review and editing, all authors; funding acquisition, J.M.S. and C.L.P. All authors have read and agreed to the published version of the manuscript.

**Funding:** This research was funded by John Templeton Foundation (grants 61513.and 61127).

**Institutional Review Board Statement:** Not applicable.

**Informed Consent Statement:** Not applicable.

**Data Availability Statement:** Data are available publicly at https://osf.io/d869y.

**Conflicts of Interest:** The authors declare no conflict of interest.

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
