# Peer review of "U.S. Federal Investment in Religiousness/Spirituality and Health Research: A Systematic Review"

_religions, doi:10.3390/rel13080725_

Round 1
Reviewer 1 Report
The authors conducted a systematic review of grants that have been funded to study religiousness/spirituality (R/S) and health. They found that a substantial amount of funding has been spent on R/S and health although the amount has been declining. I thought the manuscript was interesting and presents a useful review. I hope my comments can help improve the manuscript.
Introduction
“However, the quality of research on R/S and health has generally been low, with most studies using weak research designs (e.g., cross-sectional observational) and failing to include or control for confounding variables”
This statement seems a bit harsh. I would rephrase by noting certain limitations while also recognizing that all methods have their own strengths and weaknesses.
Search strategy
Why was the search stopped at 2018?
When were the impact factors of the journals gathered?
How does the funding of the R/S grants compare to the funding of NIH grants more broadly? Is the average R/S grant similar to the average NIH grant?
Author Response
- The authors conducted a systematic review of grants that have been funded to study religiousness/spirituality (R/S) and health. They found that a substantial amount of funding has been spent on R/S and health although the amount has been declining. I thought the manuscript was interesting and presents a useful review. I hope my comments can help improve the manuscript.
We appreciate the kind words and helpful feedback.
- Introduction: “However, the quality of research on R/S and health has generally been low, with most studies using weak research designs (e.g., cross-sectional observational) and failing to include or control for confounding variables.” This statement seems a bit harsh. I would rephrase by noting certain limitations while also recognizing that all methods have their own strengths and weaknesses.
Thank you for this comment; we agree and have modified the language to be less critical and more explanatory of our point.
- Search strategy: Why was the search stopped at 2018? When were the impact factors of the journals gathered?
The search was stopped in 2018 to coincide with the funding period of the project and avoid data from a partial year. Unfortunately, the US government discontinued the Federal Reporter and the system is no longer available for searching or updating. The impact factors of the journals were gathered in 2020.
- How does the funding of the R/S grants compare to the funding of NIH grants more broadly? Is the average R/S grant similar to the average NIH grant?
We have another manuscript that describes the comparison of R/S grants to conceptually similar grants on other constructs (e.g., social support) within the NIH portfolio. To compare R/S grants to all NIH grants more broadly would be a herculean undertaking, but we compared the funding per NIH R01 grant for R/S and health to that of a comparable psychosocial resource, social support, and found almost the same dollar amount. This could be because federal grants are capped at certain amounts and grant applications usually budget right up to that limit.
5 Year R01s
R/S: avg. amount $2,703,457 (N=34)
SS: avg. amount $2,739,209 (N=62)
Reviewer 2 Report
Overall, this is an important and informative review. It will be an important contribution to the field. I have only a few suggestions for improvement. First, I am concerned that the only search term used for health outcomes was "and health." For example, would some research on R/S and depression be excluded if "health" is not a search term associated with it? Second, it would be valuable to include a chart listing specific health outcomes that have been funded (not just the broad categories used). Third, while this paper is about R/S broadly, I am unsure how it ties into gratitude. Is would highly recommend adding something on gratitude to God to make it more relevant to the special section. Gratitude to God is a novel area of research so adding it to the body of the review will not likely be possible but perhaps you could note in the future directions that since gratitude is so clearly linked to positive health outcomes, gratitude to God could be an interesting area for future research.
Author Response
- Overall, this is an important and informative review. It will be an important contribution to the field. I have only a few suggestions for improvement.
We appreciate the enthusiasm for our work and helpful suggestions.
- First, I am concerned that the only search term used for health outcomes was "and health." For example, would some research on R/S and depression be excluded if "health" is not a search term associated with it?
We wanted to provide as broad of a search strategy as possible and so we anticipated that “health” would be sufficiently comprehensive to include outcomes focused on both physical and mental health and we did not limit our search terms to the title or abstract only. What that means is that even if a project does not specifically reference health in the title or abstract, if it had a mental health outcome (e.g., depression, anxiety), then these outcomes would have been indexed as part of the project terms and thus captured by our search strategy. For example, the NIH-funded studies “Meaning-centered Group Psychotherapy in Advanced Cancer” (PI: Breitbart) and “Building Evidence for Effective Palliative/End of Life Care for Teens with Cancer” (PI: Lyon) were included in our review. Neither project specifically references “health” in the abstract or title but include “depression” in the abstracts and were captured by our search strategy. We added this explanation to our manuscript.
- Second, it would be valuable to include a chart listing specific health outcomes that have been funded (not just the broad categories used).
The health outcomes assessed were not always fully described in the abstracts, and when they were, they were quite diffuse and included a wide array of objective (e.g., urine toxicology, blood pressure, STI screenings, 6 min walk) and subjects (e.g., physical, emotional, social, spiritual wellbeing) outcomes. Given the variety and frequency, summarizing the categories of disease seemed to be the most meaningful and potentially interpretable approach.
Thus, to better address this issue, we revisited our original health categories to determine if we could provide more meaningful information for those groups. We edited Table 1 to include additional subcategories for studies in our “Other” group to identify those focused on HIV/AIDS, multiple chronic conditions, and cardiovascular disease. We also provided examples for the “Other” categories not specifically described in the initial submission (e.g., amyotrophic lateral sclerosis, asthma, infertility, end-stage kidney disease, obesity).
- Third, while this paper is about R/S broadly, I am unsure how it ties into gratitude. Is would highly recommend adding something on gratitude to God to make it more relevant to the special section. Gratitude to God is a novel area of research so adding it to the body of the review will not likely be possible but perhaps you could note in the future directions that since gratitude is so clearly linked to positive health outcomes, gratitude to God could be an interesting area for future research.
We apologize for the earlier omission of the quantitative information on gratitude and other religion-relevant virtues and now include it as well as describe future directions for these variables in the discussion.